# Land Surface Temperature Variation in Response to Land Use Modes Changes: The Case of Mefou River Sub-Basin (Southern Cameroon)

Valentin Brice Ebodé [1,2]

1 Department of Geography, University of Yaounde 1, Yaounde P.O. Box 755, Cameroon; ebodebriso@gmail.com; Tel.: +237-694426200
2 International Joint Laboratory DYCOFAC, IRGM-UY1-IRD, Yaounde P.O. Box 4110, Cameroon

**Abstract:** Land surface temperature (LST) estimation at the river sub-basin level is crucial for developing land use planning at the basin scale and beyond. The main goal of this study was to analyze LST variations in response to land use mode (LUM) changes in the Mefou River sub-basin (Southern Cameroon) using geospatial techniques. To achieve this goal, we used Landsat 7 Enhanced Thematic Mapper Plus (2000 and 2010) and Landsat 8 Operational Land Imager (OLI)/Thermal Infrared Sensor (TIRS) data for 2020. We also used air surface temperature data from the Climatic Research Unit (CRU) to validate the LST. Our results reveal that between 2000 and 2020, the Mefou watershed recorded significant changes in LUMs, which were mainly manifested by an increase in impervious areas (IAs) (buildings and roads (+10%); bare soils and farmlands (+204.9%)) and forest reduction (−31.2). This decrease in the forest was also reflected by a reduction in NDVI values, the maximum of which went from 0.47 in 2000 to 0.39 in 2020. Contrary to the forest area and the NDVI values, the LSTs of the investigated basin increased over the period studied. There is a strong negative correlation between LST and NDVI. In general, high LSTs correspond to low NDVI values. For the years 2000, 2010 and 2020, the links between these two variables are materialized by respective correlation coefficients of −0.66, −0.74 and −0.85. This study could contribute to understanding the impact of LUM changes on the local climate, and could further provide assistance to policymakers in regard to land use planning and climate change mitigation strategies.

**Keywords:** land surface temperature; land use modes; normalized difference vegetation index; Mefou

## 1. Introduction

The combined effect of rapid population growth and the overexploitation of natural resources increases the problem of LUM changes [1–3]. Changes in LUM have been shown to significantly influence climate systems in general [4–8], and local climate in particular [9].

The exploitation of natural resources and increases in human activities have contributed to increases in the LST, which in turn worsen the dynamics of LUMs [4,10]. LST is determined by the types of land elements and the rate of radiated energy emitted from the ground surface [11]. LST represents the cumulative effects of LUM changes, rainfall variations, and socio-economic development [12]. Anthropogenic activities have a large share of responsibility in the deterioration of the environment. For example, the study by Gemes et al. [13] revealed that environmental degradation has been a major environmental problem since agricultural activities began [13]. In developing countries such as Cameroon, overgrazing, deforestation and unplanned human settlements are some major environmental issues that significantly affect sustainable development.

Substantial research works have been conducted by several authors [4,7], indicating the effects of LUM changes on LST. Feng et al. [4] showed that LST has been increasing and warming areas have been expanding since 1996, especially in the Su–Xi–Chang urban agglomeration. The mean LST in Su–Xi–Chang has increased from less than 30 °C in 1996

to greater than 31 °C in 2004, rising to about 33 °C in 2016. The projection suggests that LST will reach about 35 °C in 2026. Their results also suggest that mean LST increased by 2 °C per decade in this highly urbanized area between 1996 and 2026. In the study of Wang et al. [7], the LST response to LUM changes was comparatively evaluated using micrometeorological observations from a cropland site, a mixed forest site, a shrubland site, and adjacent bare land sites in the Heihe River Basin, Northwest China. The surface temperature changes were divided into contributions from changes in radiative forcing, heat resistance, evapotranspiration, soil heat flux, and air temperature based on the intrinsic biophysical mechanism (IBM) and the two-resistance mechanism (TRM). The results indicate that the IBM attribution method is more applicable than the TRM method in these arid ecosystems; the influence of different types of vegetation cover on the surface temperature exhibits temporal variance in the diurnal and seasonal time scales. The dominant biophysical components in the daytime of the growing season are evaporative cooling in the cropland paired sites and heat resistance change in the mixed forest paired sites, but these two components are both at a moderate level in the shrubland paired sites.

Although a significant number of studies have been conducted globally, knowledge related to LST in Central Africa is limited [14].

Previously, many studies [15,16] have been conducted on the impact of LUM changes on the environment in general, and on LST in particular. The majority of early studies focused on the urban center while marginalizing the rural area. Moreover, the effects of LUM changes on the LST at the sub-basin level are still unknown and require more studies. Tracking LST in response to LUM changes is crucial for informing policymakers of mitigation strategies. A clear understanding and knowledge of the impact of LUM changes on LST can be of great importance for environmental planning and management. The availability of reliable and up-to-date information on LUM changes in relation to LST is crucial for sustainable land use planning in general and microclimate change mitigation in particular. In addition, the dissemination of information on the impact of LUM changes on LST at the sub-basin level can be helpful for conservation decision-making bodies. Thus, the present study aims to quantify LST variation in response to LUM changes. The objective of this study is to evaluate the effects of LUM changes on LST in the Mefou River sub-basin for the period of 2000–2020.

## 2. Materials and Methods

### 2.1. Study Area

The study focuses on the Mefou watershed (428 km$^2$). This basin is located in South Cameroon, within the Central African sub-region, between latitudes 3°43′ N and 3°58′ N and longitudes 11°21′ E and 11°35′ E (Figure 1). It belongs to the sub-equatorial domain, with abundant annual precipitation (around 1600 mm/year) spread over four seasons of unequal importance. Two of them are dry (summer and winter) and two are rainy (spring and autumn). The studied basins are dissected by deep gullies cut into hills with convex slopes and wide marshy valleys. Their geological substratum is made up of a granito–gneissic base on which ferralitic soils (on the summits and slopes) and hydromorphic soils (in the shallows) develop. The vegetation in the area is a dense semi-deciduous forest with Sterculiaceae and Ulmaceae, and is subject to anthropogenic pressure [17].

### 2.2. Data Sources

This study used Landsat 7 Enhanced Thematic Mapper Plus (2000 and 2010) and Landsat 8 Operational Land Imager (OLI)/Thermal Infrared Sensor (TIRS) data from 2020. These Landsat images were obtained from the United States Geological Survey (https://www.usgs.gov/products/data-and-tools/realtime-data/remote-land-sensing-and Landsat; accessed on 12 July 2022) during the dry season (January and December) and are cloud-free (Table 1). All data were projected to UTM (Zone 32) and WGS 84 data. The study area lies within the 185 Path and 57 Row reference system. The acquired data were used for LUM classification. The data and their sources are presented in Table 1.

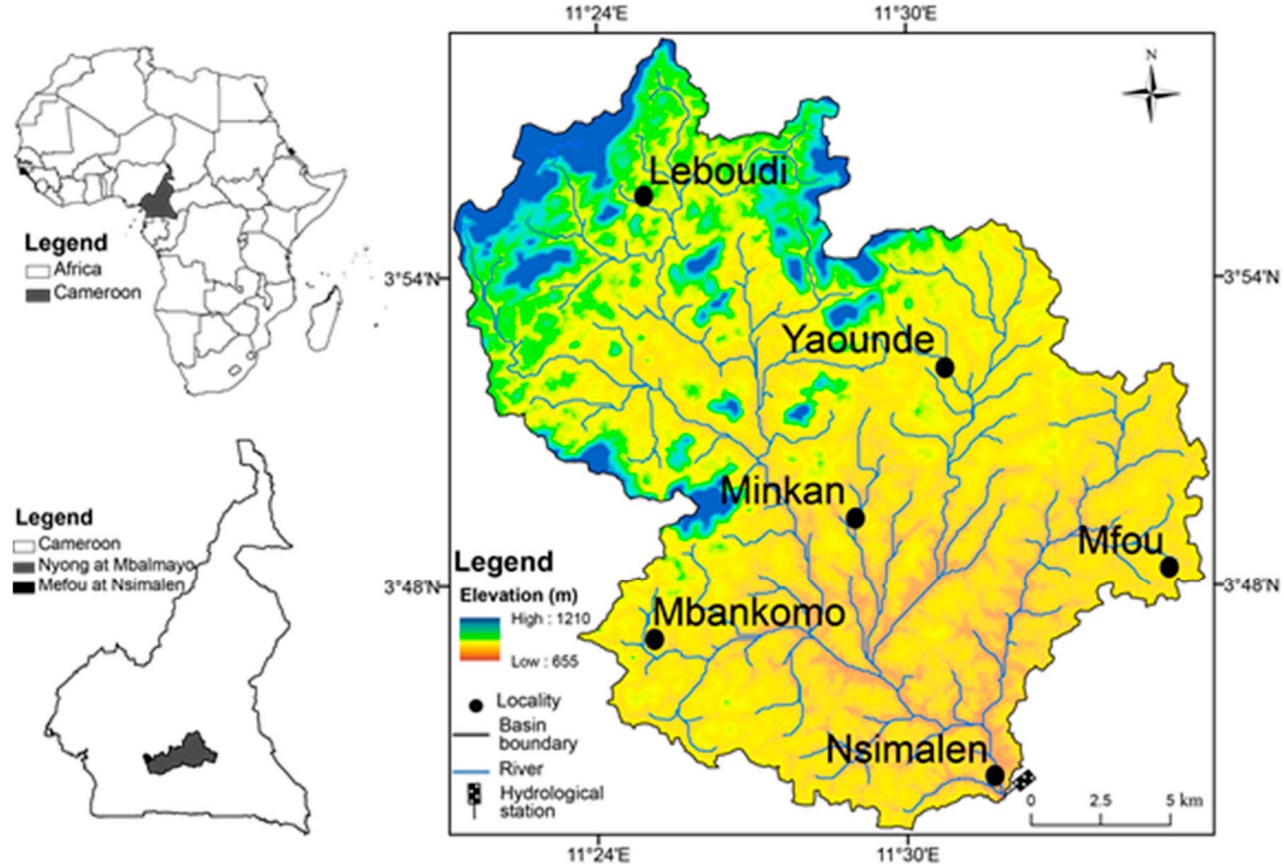

**Figure 1.** Location map of Mefou watershed at Nsimalen outlet.

**Table 1.** Remote sensing images.

| Images | Path | Row | Pixel Size | Acquisition Date | Sources |
|---|---|---|---|---|---|
| Landsat ETM+ of 2000 | 185 | 57 | 30 × 30 | January 2000 | U.S. Geological Survey |
| Landsat ETM + of 2010 | 185 | 57 | 30 × 30 | December 2010 | U.S. Geological Survey |
| Landsat OLI/TIRS of 2020 | 185 | 57 | 30 × 30 | January 2020 | U.S. Geological Survey |

The maximum and minimum air temperature data used in this study are from the CRU (Climate Research Unit) of the University of East Anglia in the United Kingdom. These data have been available since 1901 via the site https://climexp.knmi.nl/selectfield_obs2.cgi?id=2833fad3fef1bedc6761d5cba64775f0/, accessed on 12 July 2022, in NetCDF format with monthly time steps at a spatial resolution of 0.25° × 0.25°. Precipitation and temperature data from the CRU have been used to validate CMIP models in the Logone basin, which is a sub-basin of the Lake Chad basin [18].

*2.3. Land Use/Land Cover Classification*

Landsat images were classified using supervised maximum likelihood classification and Sentinel Application Platform (SNAP) software, which is made available to the general public for free by the European Space Agency (ESA) via the site https://step.esa.int/main/download/snap-download/, accessed on 12 July 2022. This enabled us to perform a diachronic analysis of the evolution of land-use in the studied basin. This operation was preceded by operations of preprocessing and the recognition of objects in the field by photography and GPS (global positioning system). Satellite image preprocessing refers to all the processes applied to raw data to correct geometric and radiometric errors that characterize certain satellite images. These errors are generally due to technical problems with the satellites and interactions between outgoing electromagnetic radiation and atmospheric

aerosols, also called "atmospheric noise". The atmospheric disturbances are influenced by various factors that are present on the day of acquisition, including weather, fires, and other human activities. They affect all the images acquired by passive satellites including Landsat 4-5-7 and 8. As the downloaded Landsat images were orthorectified, the preprocessing involved atmospheric correction of the images and reprojection into the local system (WGS_84_UTM_Zone_32N). For this, neo-channels were created to increase the readability of the data by enhancing certain properties that were less obvious in the original image, thus showing more clearly the elements of the scene. Three indices are therefore created, namely, the Normalized Difference Vegetation Index (NDVI, Equation (1)), the Brightness Index (BI, Equation (2)), and the Normalized Difference Water Index (NDWI, Equation (3)) [19,20]. These indices highlight vegetated surfaces and sterile (non-chlorophyllin) elements such as urban areas and water bodies, respectively. The formulae used in creating these indices are as follows:

$$NDVI = \frac{NIR - R}{NIR + R} \tag{1}$$

$$BI = \left(R^2 + NIR^2\right)^{0.5} \tag{2}$$

$$NDWI = \frac{NIR - MWIR}{NIR + MWIR} \tag{3}$$

where NIR is the ground reflectance of the surface in the near-infrared channel; R is the ground reflectance of the surface in the red channel; and MWIR is the ground reflectance of the surface in the mid-wave infrared channel. The use of Google Earth, as well as the spaces sampled from the GPS, made it possible to identify with certainty the impervious areas (buildings, savannas, bare soils, and crops), water bodies (large rivers, lakes and ponds) and forest (secondary, degraded, non-degraded and swampy) of each mosaic. Before the classification, the separability of the spectral signatures of the sampled objects to avoid interclass confusion was assessed by calculating the "transformed divergence" index. The value of this index is between 0 and 2. A value > 1.8 indicates a good separability between two given classes. The different classes used in this study show good separability between them, irrespective of the image considered, with indices > 1.9. The validation of the classifications obtained was carried out using the confusion matrix, making it possible to obtain treatment details to validate the choice of training plots. After validating the land use/land cover maps, the statistical and spatial differences of each class between the studied periods were evaluated.

*2.4. LST Retrieval*

LST is influenced by topography, landscape composition, land cover, urbanization, and global change [4,21]. It is affected by albedo, vegetation cover, and soil moisture [22]. The ETM + and TIRS thermal band calibration constants in this study are presented in Table 2. LST has been calculated [22] in many steps using Landsat ETM+ and the Landsat 8 Operational Land Imager.

**Table 2.** ETM + and TIRS thermal band calibration constants.

| | Constant 1-K1 Watts/(m$^2$∗ster∗μm) | Constant 2-K2 Kelvin |
|---|---|---|
| Landsat 7 | 666.09 | 1282.71 |
| Landsat 8 | 774.8853 | 1321.0789 |

2.4.1. Step I: Conversion of the Digital Number (DN) into Spectral Radiance (L)

In the present study, digital numbers were converted to at-sensor radiance values prior to calculating brightness temperature. The ETM + DN values range between 0 and 255 (Equation (4)).

$$L\gamma = \frac{LMAX\gamma - LMIN\gamma}{QCALMAX - QCALMIN} \times (DN - QCALMIN) + LMIN\gamma \tag{4}$$

where QCAL is the quantized calibrated pixel value in digital number (DN), LMIN$\gamma$ is the spectral radiance that is scaled to QCALMIN in (Wm$^2$ster$^{-1}$μm$^{-1}$), LMAX$\gamma$ is the spectral radiance that is scaled to QCALMAX in (Wm$^2$ster$^{-1}$μm$^{-1}$), QCALMIN is the minimum quantized calibrated pixel value corresponding to LMIN$\gamma$ in DN, and QCALMAX is the maximum quantized calibrated pixel value corresponding to LMAX$\gamma$ in DN = 255.

At first, digital numbers were converted from the Landsat 8 Operational Land Imager to spectral radiance, then brightness temperature was extracted from thermal remote sensing data (TRSD) [10]. In the Landsat 8 data from the radiance multiplier (ML) and radiance add (AL), the thermal infrared (TIR) band was converted into spectral radiance L$\gamma$ using the approach provided by Chander and Markhan [23] and used by Chibuike et al. [22], as indicated in (Equation (5)).

$$L\gamma = (M_L * Q_{cal}) + A_L \tag{5}$$

where L$\gamma$ is the top of the atmosphere spectral radiance (Wm$^2$ster$^{-1}$μm$^{-1}$), $M_L$ is the band-specific multiplicative rescaling factor from the metadata (RADIANCE_MULT_ BAND_x, where x is the band number), $A_L$ is the band-specific additive rescaling factor from the metadata (RADIANCE_ ADD_BAND_x, where x is the band number) and $Q_{cal}$ is the quantized and calibrated standard product pixel values (DN).

2.4.2. Step II: Conversion to Brightness Temperature

Brightness temperature and average atmospheric temperature were used to calculate LST based on land surface emissivity (Chibuike et al. [22]). The specific formula for the mono-window algorithm for retrieving LST was used (Chibuike et al. [22]). The black body temperature was obtained from the spectral radiance using Plank's inverse function. Spectral radiance values for bands 6 and 10 were converted to radiant surface temperature under assumptions of uniform emissivity using pre-launch calibration constants (Chibuike et al. [22]). The Landsat satellite images were converted from spectral radiance to a more physically useful variable. The conversion formula is presented in (Equation (6)).

$$T = \frac{K2}{\ln\left(\frac{K1}{L\gamma} + 1\right)} \tag{6}$$

where $T$ is effective at satellite temperature in Kelvin, $K2$ is the calibration constant 2, $K1$ is the calibration constant 1, and L$\gamma$ is the spectral radiance in Wm$^2$ster$^{-1}$ μm$^{-1}$.

2.4.3. Step III: Land Surface Emissivity Estimation

The land surface emissivity estimation was performed (Chibuike et al. [22]) and computed using Equation (7).

$$\varepsilon = 0.005 * P_v + 0.986 \tag{7}$$

where $P_v$ is the vegetation proportion obtained (Carlson and Ripley [24]) using Equation (8).

$$PV = \left[\frac{NDVI - NDVImin}{NDVImax - NDVImin}\right]^2 \tag{8}$$

In this study, the calculated radiant surface temperature was corrected for emissivity (Chibuike et al. [22]) using Equation (9).

$$LST = \frac{TB}{1\left(\gamma\frac{TB}{P}\right)\ln\varepsilon} \tag{9}$$

where *LST* is land surface temperature (in Kelvin), *TB* is the radiant surface temperature (in Kelvin), $\gamma$ is the wavelength of emitted radiance (10.8 μm), *P* is $h*c/\sigma(1.438*10^{-1}$ mK), h is Planck's constant ($6.26*10^{10-34}$ Js), c is the velocity of light ($2.998*10^8$ m/s), σ is Stefan Boltzmann's constant ($1.38*10^{-23}$ J K$^{-1}$), and ɛ is land surface emissivity.

Finally, the LST results from Landsat ETM+ and OLI/TIRS in Kelvin degrees were converted into Celsius degrees by subtracting 273.15. (Equation (6)).

### 2.5. Analysis of Annual Maximum and Minimum Air Temperatures

The Mann–Kendall test at the 95% significance level was used to analyze the mean annual maximum and minimum air temperatures. This test is based on the test statistic "S", defined as follows:

$$S = \sum_{i=1}^{n-1} \sum_{j=i+1}^{n} sgn(xj - xi) \tag{10}$$

where xj represents the sequential data values, n is the length of the data set, and sgn = (θ) if θ > 1, 0 if θ = 0, and −1 if θ < 0. There is no significant trend in the series analyzed when the calculated p-value is above the chosen significance level.

## 3. Results and Discussion

### 3.1. Changes in Land Use Modes

A diachronic analysis of the classifications carried out from the processing of Landsat satellite images from three dates (2000, 2010 and 2020) shows a significant change in land use modes in the Mefou watershed (Figure 2). These changes are essentially reflected in an increase in IAs to the detriment of the forest towards the northwest of the basin (Figure 3). Between 2000 and 2020, buildings and bare soils increased by +10% and +204.9%, respectively (Table 3). Most of these changes occurred between 2000 and 2010. Over this interval, buildings and bare soils increased by +7.6% and +101.8%, respectively. Their increases between 2010 and 2020 were less. They are +2.3% and +51.1%, respectively (Table 3). In the case of forests and water bodies, the decreases noted are −31.2% and −14.6%, respectively (Table 3). Concerning the forest, the rates of decrease recorded between 2000 and 2010 (−16.5%) and between 2010 and 2020 (−17.6%) are very similar. For water bodies, on the other hand, most of the decline observed over the entire period studied occurred between 2010 and 2020 (Table 3). Some authors in Central Africa [25–28], West Africa [29,30] and elsewhere [31,32] have made similar observations relating to changes in land use modes in these sub-regions.

**Table 3.** Evolution of the main land use modes in the Mefou watershed during the study period.

| Land Use Modes | Area Occupied in the Basin (km$^2$) | | | 2000–2010 | | 2010–2020 | | 2000–2020 | |
| --- | --- | --- | --- | --- | --- | --- | --- | --- | --- |
| | **2000** | **2010** | **2020** | **km$^2$** | **%** | **km$^2$** | **%** | **km$^2$** | **%** |
| Built and roads | 118.1 | 127.1 | 130 | 9.0 | 7.6 | 2.9 | 2.3 | 11.9 | 10 |
| Forest | 273.3 | 228.1 | 188 | −45.2 | −16.5 | −40.1 | −17.6 | −85.3 | −31.2 |
| Water | 0.8 | 0.8 | 0.7 | 0.0 | −2.4 | −0.1 | −12.5 | −0.1 | −14.6 |
| Bare soils and farm lands | 35.7 | 72.1 | 109 | 36.3 | 101.8 | 36.8 | 51.1 | 73.2 | 204.9 |

### 3.2. Normalized Difference Vegetation Index (NDVI) Analysis

The spatial analysis of the NDVI results shows that the NDVI values considerably decreased between 1990 and 2020 in the investigated basin (Figure 4). During the years 2000, 2010 and 2020, the maximum values of NDVI were 0.47, 0.44 and 0.39, respectively (Figure 4). These figures reflect the reduction in green spaces in the basin. Yang et al. [33] made a similar observation in their study in China. The results relating to the NDVI values during the years studied show that the forest which covered the surroundings of Yaounde city (Mbankomo, Mfou and Nsimalen) in 2000 have been gradually replaced by buildings, roads, bare soils, and crops (Figure 2).

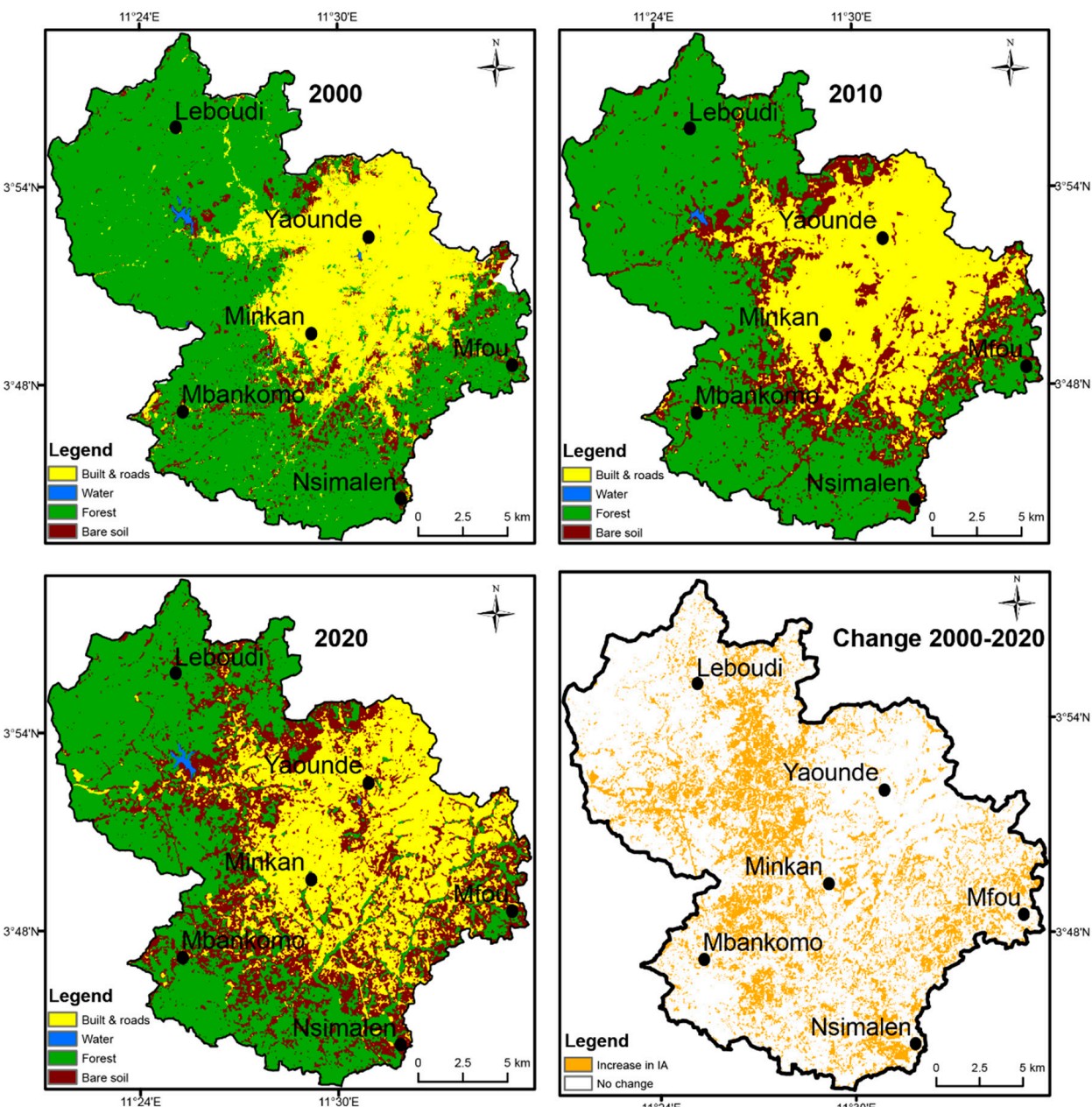

**Figure 2.** Changes in the spatial distribution of the main land use modes in the Mefou basin between 2000 and 2020.

The Landsat image analysis indicates that NDVI and LST have a strong relationship. Low NDVI values correspond to high LSTs. Statistical correlation analysis demonstrates that NDVI has a strong negative correlation with "r" values of −0.66, −0.74, and −0.85 for the years 2000, 2010, and 2020 (Figure 5). Some authors [16,34] made similar observations. According to them, the vegetation land cover class has the potential to cool the environment and is very effective in climate change mitigation strategies. Vegetation coverage could decrease the surface and air temperature by providing shade, which saves land surfaces from the direct heat of sunlight [34,35]. Conversely, built areas, roads, farmland, and bare soils have high LSTs.

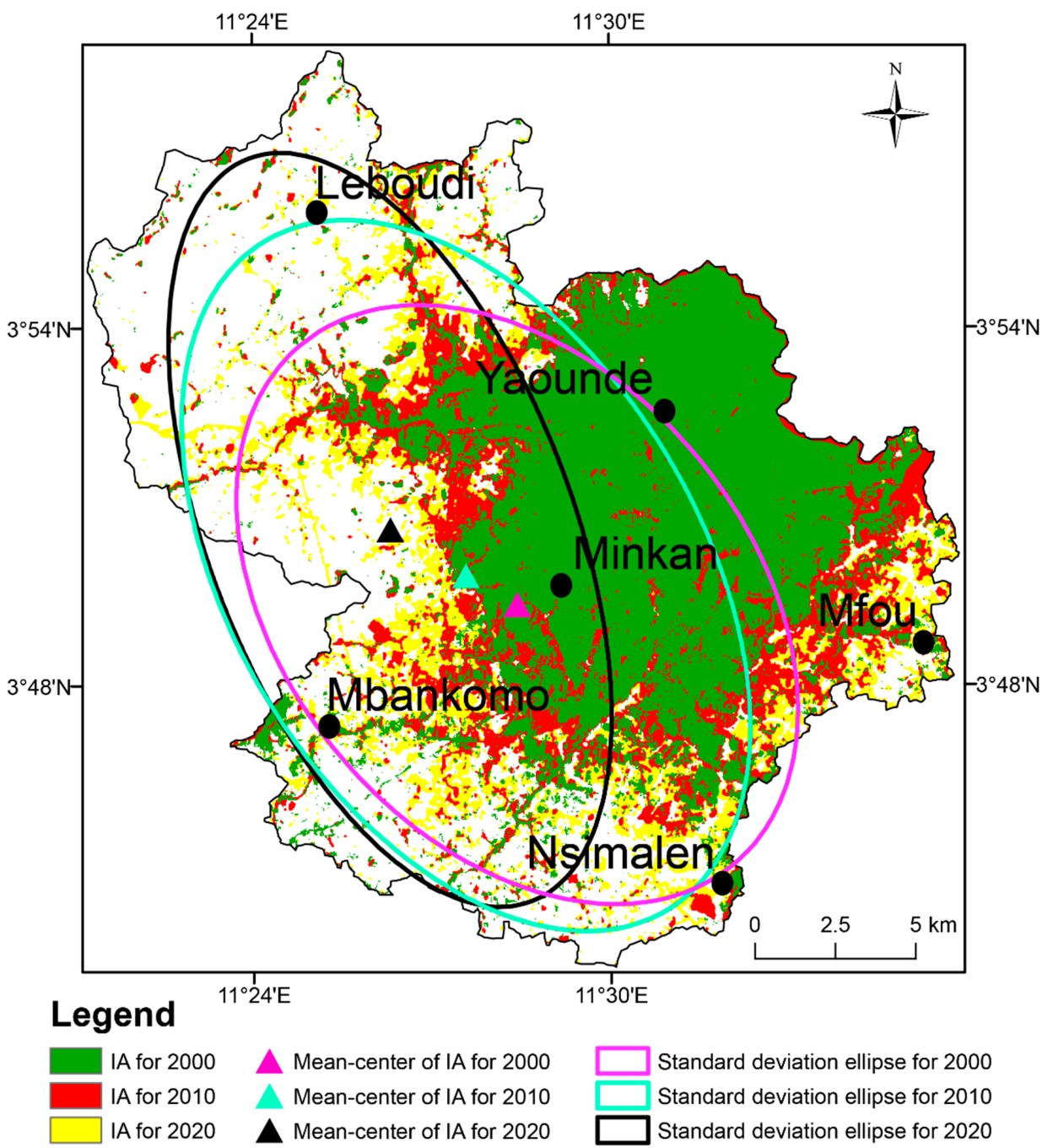

**Figure 3.** Evolution of impervious areas (IAs) in the Mefou basin between 2000 and 2020. IAs include built areas, roads, bare soils and farmlands.

### 3.3. LST Variation in Response to LUM Dynamics

LST increased in the Mefou watershed over the interval studied (Figure 6). During the years 2000, 2010, and 2020, the maximum LSTs values were 34 °C, 49 °C, and 54 °C, respectively. The minimum LSTs values were 17 °C, 18 °C, and 45 °C (Figure 6). For the maximum LST, the increases noted between 2000 and 2020 were +20 °C. Most of this increase took place between 2000 and 2010 (+15 °C). In the case of minimum LST, the increase noted between 2000 and 2020 was +28 °C. This increase mainly occurred between 2010 and 2020.

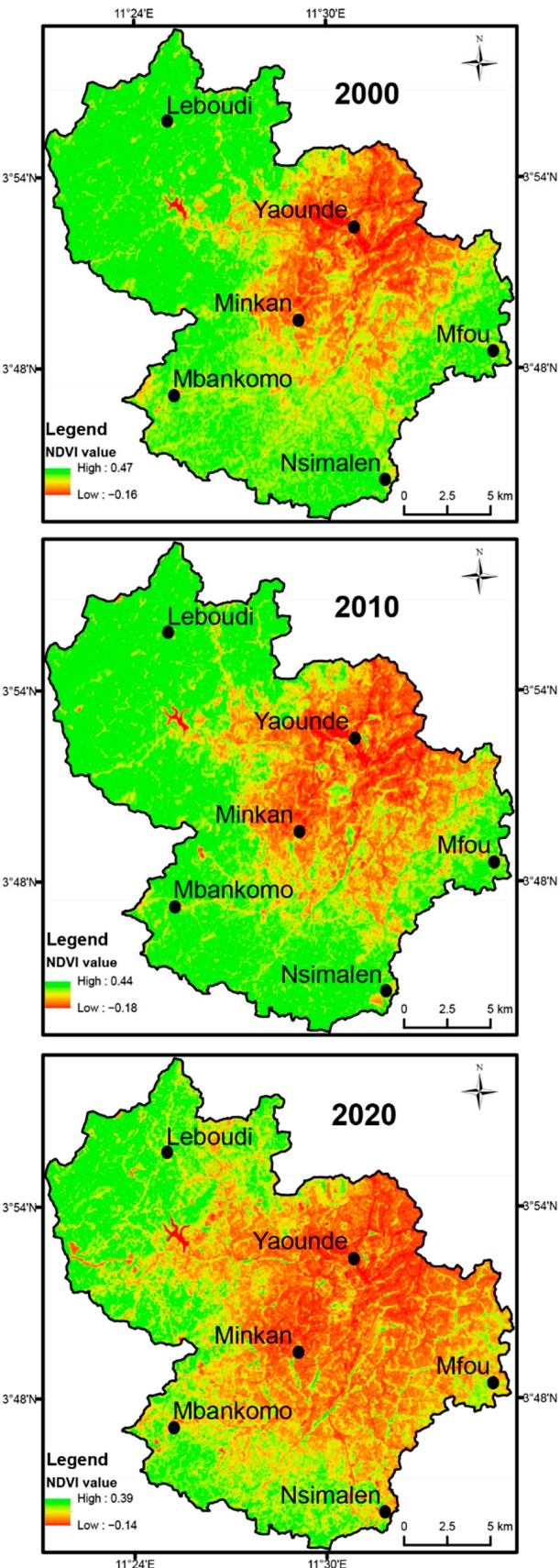

**Figure 4.** NDVI maps from 2000 to 2020.

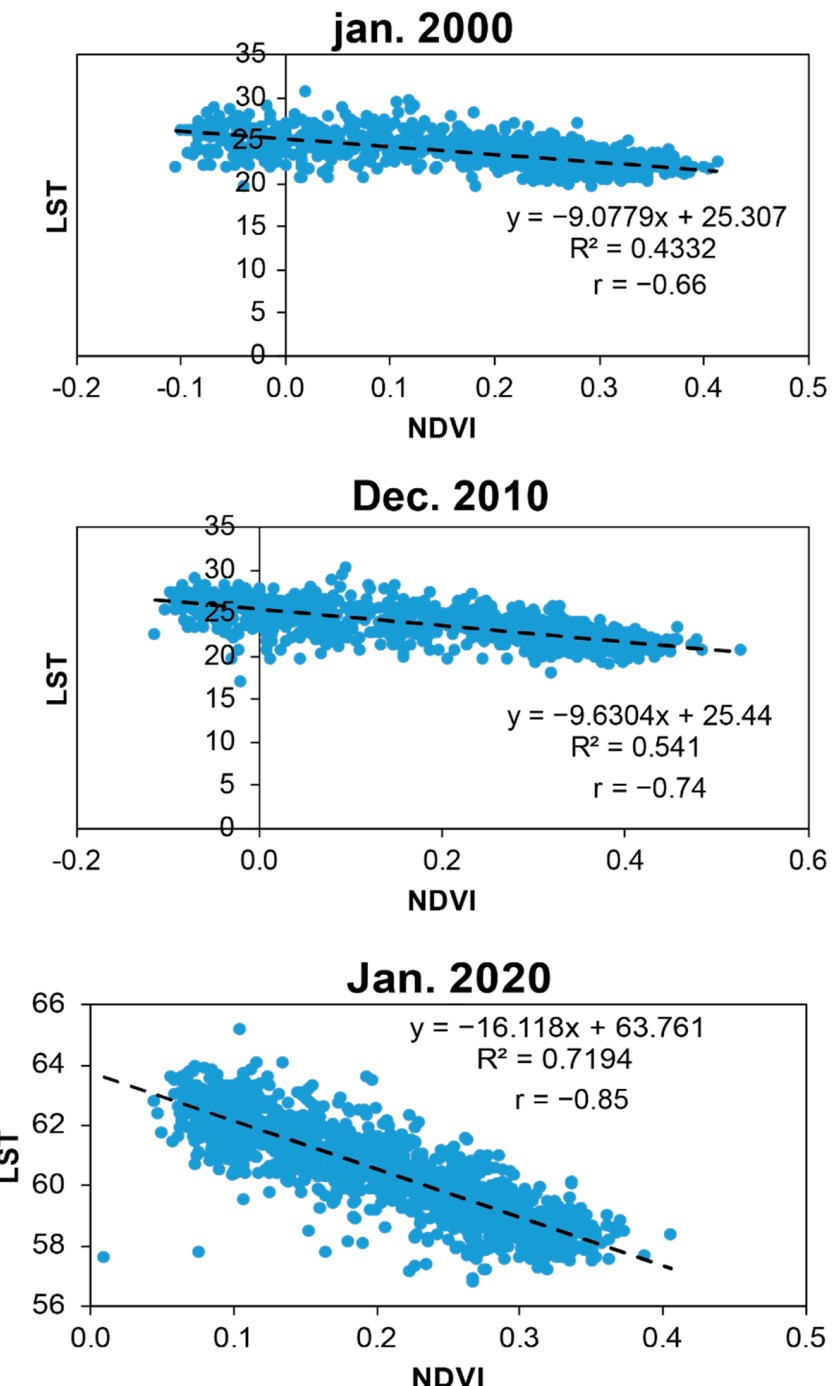

**Figure 5.** Linear regression scatter plot of LST and NDVI in 2000, 2010, and 2020.

In general, low LST is observed in forested areas and high LST is observed in areas covered by buildings and bare soils. In 2000, 2010, and 2020, for example, 94%, 91%, and 98.7% of the lowest LSTs (17–22 °C, 18–21 °C, and 45–47 °C, respectively) were observed in areas covered by forest. During these same years, 81.6%, 87.5%, and 88.4% of the highest LSTs (28–34 °C, 27–49 °C, and 51–55 °C, respectively) were observed in the spaces covered by buildings (Table 4). Other studies [36,37] have already shown that the highest and lowest LSTs are generally observed in bare spaces and spaces covered with vegetation, respectively.

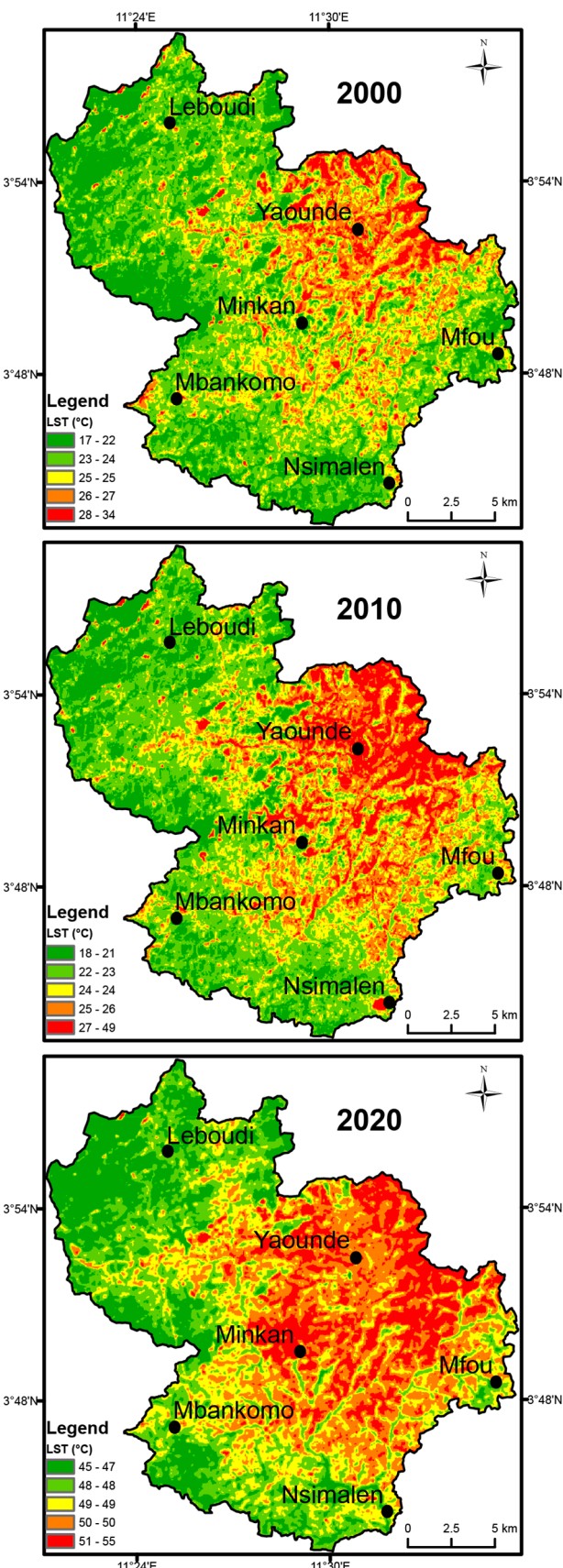

**Figure 6.** LST maps from 2000 to 2020.

**Table 4.** Statistical links between LST and LUM.

| LST Year | LST Classes | Area | | % of LST Class Observed in Each LUM (%) | | | |
|---|---|---|---|---|---|---|---|
| | | km$^2$ | % | Built and Roads | Forest | Water | Bare Soils and Farm Lands |
| 2000 | 17–22 | 96.6 | 22.6 | 5.1 | 94 | 0.5 | 0.4 |
| | 23–24 | 161.0 | 37.6 | 12.8 | 83.9 | 0.2 | 3.1 |
| | 25–25 | 94.2 | 22.0 | 41.7 | 44.3 | 0 | 14 |
| | 26–27 | 55.7 | 13.0 | 73.2 | 11.1 | 0 | 15.6 |
| | 28–34 | 20.6 | 4.8 | 81.6 | 3.9 | 0 | 14.6 |
| 2010 | 18–21 | 67.8 | 15.8 | 6.5 | 91.7 | 0 | 1.8 |
| | 23–24 | 152.6 | 35.7 | 7.5 | 83.1 | 0.2 | 9.2 |
| | 24–24 | 93.5 | 21.8 | 25.2 | 41.8 | 0 | 32.9 |
| | 25–26 | 69.9 | 16.3 | 64.9 | 6.9 | 0 | 28.2 |
| | 27–49 | 44.1 | 10.3 | 87.5 | 0.9 | 0 | 11.6 |
| 2020 | 45–47 | 76.6 | 17.9 | 0 | 98.7 | 0.9 | 0.4 |
| | 48–48 | 99.2 | 23.2 | 1.3 | 86.7 | 0 | 12 |
| | 49–49 | 94.2 | 22.0 | 17.8 | 27.3 | 0 | 54.9 |
| | 50–50 | 101.1 | 23.6 | 61.3 | 1.6 | 0 | 37.1 |
| | 51–55 | 56.9 | 13.3 | 88.4 | 0 | 0 | 11.6 |

*3.4. LST Validation Results with Respect to Air Temperature*

Minimum and maximum air temperatures increased in the Mefou watershed between 2000 and 2020. This increase is statistically significant for minimum temperature, with a *p*-value below the significance level of 0.05 (Figure 7). This implies that the increase noted for the minimum temperature is greater than that of the maximum temperature. We also note that the increase in minimum temperature is greater over the period 2010–2020 (Figure 7).

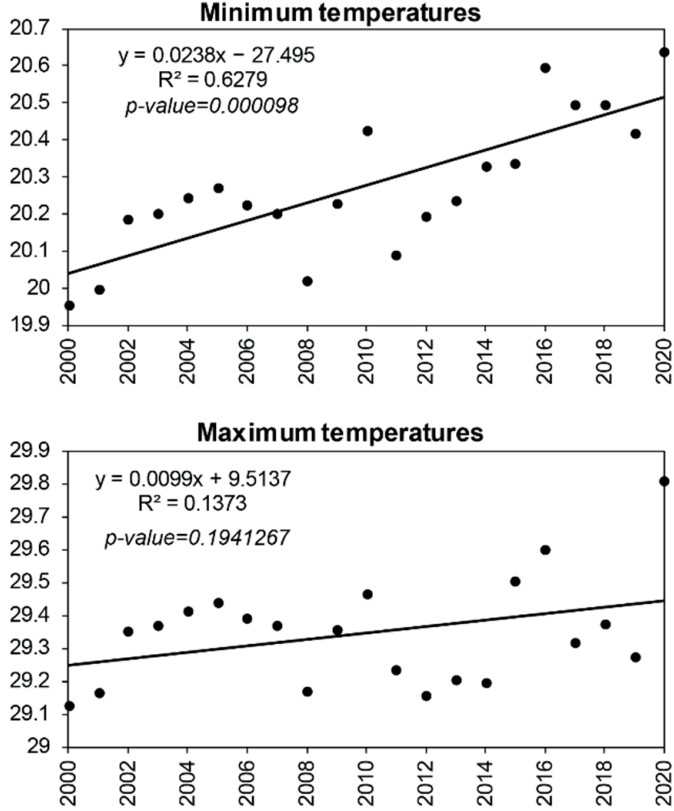

**Figure 7.** Evolution of minimum and maximum air temperatures between 2000 and 2020.

There is a strong concordance in the evolution of LST and air temperature in the Mefou watershed, which suggests that there is a link between them. We note in both cases an increase in maximum and minimum temperatures, a greater increase in minimum temperature compared to maximum temperature, and a greater increase in minimum temperature during the period 2000–2010. A similar finding was reported by Tafesse and Suryabhagavan [16] around the Adama Zuria district in Ethiopia.

## 4. Conclusions

The goal of this study was to analyze LST variations in response to LUM dynamics in the Mefou River sub-basin using geospatial techniques (Southern Cameroon). For this, we used Landsat satellite images and CRU air temperature data. Between 2000 and 2020, the Mefou watershed recorded significant changes in LUM, which were mainly manifested by an increase in IAs (buildings and roads (+10%); bare soils and farmlands (+204.9%)) and forest reduction ($-31.2$). This decrease in the forest is also reflected in a reduction in NDVI values, whose maximum values went from 0.47 in 2000 to 0.39 in 2020. Contrary to the forest area and the NDVI values, the LST of the investigated basin increased over the period studied. The maximum and minimum values increased from 34 °C to 54 °C and from 17 °C to 45 °C, respectively. There is a strong negative correlation between LST and NDVI. In general, high LSTs correspond to low NDVI values. For the years 2000, 2010, and 2020, the links between these two variables are materialized by the respective correlation coefficients of $-0.66$, $-0.74$, and $-0.85$. Based on our results, it is recommended to increase the campaigns of afforestation and reforestation programs to minimize unexpected increases in LST in the study area and beyond. Further studies should be conducted, incorporating additional factors for a better understanding of the effects of LUM dynamics on LST for mitigation strategies.

**Funding:** This research received no external funding.

**Institutional Review Board Statement:** Not applicable.

**Informed Consent Statement:** Not applicable.

**Data Availability Statement:** All relevant data are included in the paper.

**Conflicts of Interest:** The author declares no conflict of interest.

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
