# Peer review of "Land Surface Temperature Variation in Response to Land Use Modes Changes: The Case of Mefou River Sub-Basin (Southern Cameroon)"

_sustainability, doi:10.3390/su15010864_

Round 1
Reviewer 1 Report
The research is of limited value
The innovation of the paper is not too high.
This kind of research is very common.
But it can be published after major modification.

Author Response
Cover Letter
Manuscript ID: Sustainability-2035719
Type: Article
Title: Land surface temperature variation in response to land use modes changes: a case of Mefou river sub‑basin (Southern Cameroon)
Author: Valentin Brice Ebodé
Reviewers of this work have a number of concerns about it.
Reviewer 1 asked why only images from three years were used for processing. He also wondered why we have not retained all the years of the study period (21 years) for the treatments. LANDSAT satellite images of sufficient quality for processing (haze-free and cloud-free) are not available for all years, even in the best-monitored regions of the world (in the USA), even less in the region studied (Central Africa). There are even very long periods during which they are not available in the region studied. The choice of three relatively spaced images to conduct our study is therefore reasonable, given the difficulty of acquiring satellite images. Several studies dealing with this question before this one have proceeded similarly.
The choice of images for January 2000, December 2010, and January 2020 (reviewer 1 concern) is because these months belong to the same season in the region studied (long dry season). For such studies, it is always good to use images from the same season, because the temperatures are relatively close, and the phonological stage of the vegetation is also practically the same. In addition to this, the season chosen corresponds to the period for which the satellite images are of better quality (containing little cloud and mist). Failing to have the images of the same date, the ideal would have been to have the images of the same month for all the years, but this proved to be impossible, given the unavailability of the images already mentioned above. Hence the choice of images from the same season.
Reviewer 1 asked, where the points in subfigures a, b, and c of Figure 5 come from. These figures show the linear correlations between LSTs and NDVIs over the three years of the study. This is done relatively simply in Excel, by crossing the series of the two variables whose correlation is being studied.
Reviewer 1 asked if rising temperatures are related to vegetation. Of course, the rise in temperatures noted in this study is concomitant with a decrease in vegetation. We could therefore consider that the two are linked, especially since hundreds of studies before this one have already shown that vegetation helps to lower the temperature. Its decrease can therefore be the cause of an increase in temperature. This is also the reason why it is generally warmer in the city than in the countryside (in the tropical regions that I know better). In this study, the analysis of the evolution of air temperatures just aims to further consolidate what has been observed for LSTs. We have demonstrated an increase in LSTs. It was therefore quite normal to interrogate the evolution of air temperatures to see if they evolve in the same direction, given the fact that LSTs and air temperatures are linked. In this study, it was observed that air temperatures and LSTs evolve in the same direction (increase), which is quite logical.
Reviewer 1 suggested including information on soil porosity, grain size, etc. in this article. He has also proposed two articles that deal with these points on which he recommends that we rely. We regret not being able to incorporate these observations into this work. In all humility, we believe that this work does not fit very well with these points. This study deals with the links between LSTs and changes in land use patterns. It does not deal with soil properties.
Reviewer 2 asked to check some things in the text (equation 4, figures 3 and 4, etc.). All these things have been carefully checked, the calculations have been redone, and it seems that this information does not contain any problems. He also noted that the results of the Mann-Kendall test were not presented. These results have been presented and are included in subsection 3.4.
Reviewer 3 requested not to use the "We" in this work, given that it has a single author. In research, even when you are a single author, it is always better to use "We" rather than "I".
Reviewer 3 also asked how LSTs vary between day and night. This aspect of things cannot be assessed in this work, given the fact that this work relies on satellite images for the study of LSTs. Given that satellite images are generally taken during the day, it is practically impossible to address these points based on the data retained for the study.
Reviewer 3 asked why he chose supervised classification rather than another method (Random forest supervised image classification). It is unanimously accepted that supervised classification is one of the best methods of processing satellite images that exist. We, therefore, chose it for this reason, but also because it is what we master. To obtain more precise results, we have calculated indices (NDVI, BI, and NDWI) which allow us to separate as finely as possible the different land use modes. Apart from the NDVI, the use of the other two indices is limited solely to classification operations. We bring this precision because reviewer 3 asked why to have calculated these two other indices whereas we only used the NDVI.
Reviewer 3 asked for a comparison (differences and similarities) between our results and those of pre-existing studies. The discussions do indeed appear in this work following the presentation of the results. In the "Results and Discussion" section, each time we present a result, we immediately discuss this result with those of the studies already carried out on the question, when possible. It is underlined in green in this new version of the manuscript.

Reviewer 2 Report
This paper studies the land use change and LST change caused by land use change in Mefou river sub basin, which has certain reference significance for us to understand the trend of land use change and ecological environment in this region.However, the processing of research data and the presentation of scientific problems are still lacking.The main questions are as follows:
(1)I am very interested in the impact of land use change on land surface temperature change in this area, but the author is not specific about scientific issues. A clearer statement needs to be proposed by the author.Please revise line 44-50 carefully regarding the presentation of scientific questions
(2)This part should be further processed and the land use treatment method should be simplified. Because this is known to most scientific researchers. Line 93.
(3) The formulation and layout of the formula are problematic. Line 148.
2.3 Land‑Use/Land‑Cover Classification and 2.3 LST Retrieva ? Please revise carefully, many details of the article are wrong
(4)The expression in Figure 3 is not clear. Line 220-221
(5) For land surface coverage, cloud, rain and snow have higher reflection effect in visible light band than near infrared band, so NDVI is negative. The NDVI of rock and bare soil is generally 0. Places with vegetation cover are generally greater than 0. Figure 4 looks wrong. Line 233-234.
(6)There seems to be no description of Mann-Kendall in the result part of the paper.
(7)The conclusion of this paper contains part of the discussion, but this is not sufficient, please expand the discussion.
Author Response

(The authors gave the same response as above.)

Reviewer 3 Report
The manuscript entitled "Land surface temperature variation in response to land use modes changes: a case of Mefou river sub‑basin (Southern Cameroon)" aims at establishing a relationship between LST variation and land use modes (e.g., NDVI). To do this, multitemproal Landsat datasets accompanied by ground truth temperature data were used. This study is relevant and worth investigating for land management actions. The manuscript is well written and structured; however, some modification is required before it is ready for publication.
Comments and suggestions:
Abstract:
- Line 12: " To achieve this goal, We"; I can only see one author? Who is "We"? Please refine throughout the manuscript.
- Line 14: Please check the sentence.
Introduction:
- More is required on reviewing available techniques and Satellite derived products in studying land surface temperature. Besides, how LST varies between day and night times?
2. Materials and Methods
2.3. Land-Use/Land-Cover Classification
-This subsection should be "Land use land cover (LULC) classification"
- Why Maximum likelihood classification? Some authors recommend "Random forest supervised image classification"?
- Please provide references for the indices separately.
- Line 113-114: Three vegetation indices have been calculated, except for the NDVI, it is not clear how the BI and NDWI were used in this study. Please clarify.
3. Results and discussion
Major comment:
1- The introduction doesn't provide adequate review on the previous studies which have employed RS and geospatial techniques in similar studies.
3- The Discussion section should emphasize on what is new/different about your results and how they coincide with other works. More is required in drawing parallel with other studies. In addition, the discussion sections are brief and without depth, adding a little depth to these sections in explaining the possible deriving factors in the correlation analysis is necessary.

Author Response

(The authors gave the same response as above.)

Reviewer 4 Report
The author can find the comments herewith attached.

Author Response

(The authors gave the same response as above.)

Round 2
Reviewer 1 Report
The author has made some modifications and supplements, and the paper has made some improvements.
Some minor modifications and additions are needed.

Author Response
Almost all the reviewers of this article acknowledged that the paper had improved after incorporating the corrections proposed in their previous reports. They proposed some slight modifications, some of which are already in the manuscript. Their comments have been incorporated in full.
Reviewer 2 Report
It is hoped that the author can further explore the mechanism and interaction of ecological environment change and land use evolution in the region
Author Response

(The authors gave the same response as above.)

Reviewer 3 Report
2. Materials and Methods
- Please provide references for the indices separately.
Author Response

(The authors gave the same response as above.)

Reviewer 4 Report
All comments suggested in my last report not included in the new version.
Author Response

(The authors gave the same response as above.)
